# Differentiating Granger Causal Influence and Stimulus-Related Information Flow

## Abstract

Information flow is becoming an increasingly popular term in the context of under-standing neural circuitry, both in neuroscience and in Artificial Neural Networks. Granger causality has long been the tool of choice in the neuroscience literature for identifying functional connectivity in the brain, i.e., pathways along which information flows. However, there has been relatively little work on providing a fundamental theory for information flow, and as part of that, understanding whether Granger causality captures the intuitive direction of information flow in a computational circuit. Recently, Venkatesh et al. [2019] proposed a theoretical framework for identifying stimulus-related information paths in a computational graph. They also provided a counterexample showing that the direction of greater Granger causal influence can be opposite to that of information flow [Venkatesh and Grover, 2015]. Here, we reexamine and expand on this counterexample. In particular, we find that Granger Causal influence can be statistically insignificant in the direction of information flow, while being significant in the opposite direction. By examining the mutual- (and conditional-mutual-) information that each signal shares with the stimulus, we are able to gain a more nuanced understanding of the actual information flows in this system.

## 1 Introduction

Information flow is starting to gain importance in both neuroscience and artificial intelligence for understanding biological and artificial neural networks. For instance, several works have sought to gain an intuition for how deep neural networks operate by examining how information propagates through these networks [Tishby et al., 2000, Goldfeld et al., 2019, Yu et al., 2018, Tax et al., 2017]. At the same time, numerous works seek to understand the brain by understanding how information flows in biological neural circuits [Almeida et al., 2013, Brovelli et al., 2004, Bar et al., 2006, Greenberg et al., 2012, Lalo et al., 2008]. Both these areas have seen an increased use of information-theoretic tools for examining information flow. In the context of the brain, Granger causality and its derivatives—including Transfer Entropy and Directed Information—have been used extensively to understand functional relationships between different areas of the brain. On the other hand, analyses of neural networks have typically entailed the use of mutual information. In what follows, we consider both biological and artificial neural networks to be instances of neural circuits that can be modeled in the form of graph of interconnected nodes, with transmissions on edges [Venkatesh et al., 2019].

Contrast the following two interpretations of the notion of "information flow", both prevalent in the neuroscience literature: (i) the first refers to "information" in the abstract, and indicates that one part of a neural circuit influences another; (ii) the second refers to some very specific information, for example, information about a stimulus in a neuroscientific experiment, or information about two classes in an ANN. In this paper, we argue that the use of Granger Causality-based tools in neuroscience should be restricted solely for understanding the first variety of "information flow"

mentioned above. In order to make inferences on stimulus-related information flows, neuroscience should take after the field of AI and use information-theoretic tools (or approximations thereof), computed between the stimulus and the neural activity of interest. We make this point by discussing a counterexample based on a feedback communication system, which shows that Granger causal influence can be greater in a direction opposite to that of information flow. While this example has been presented before by Venkatesh and Grover [2015], they leave several questions unanswered: the authors only *compare* feedforward and feedback Granger-causal influences, and do not provide a statistical or computational analysis to back up their claims. Furthermore, they provide no immediate solution that identifies the correct flows of information in this system. In a later paper [Venkatesh et al., 2019], despite constructing a framework for analyzing information flow, the authors do not define a quantitative measure for information flow, or provide satisfactory resolution to this issue: the notion of "derived information" they define is, at best, cumbersome to apply in this setting. By undertaking a computational and statistical study of this example, we address both these drawbacks of the aforementioned works.

Granger causality, along with its derivatives, is known to have several shortcomings, which have been discussed at length previously. These criticisms have largely been associated with the fact that Granger causality does not capture true causal influence [Pearl, 2009], or that it may provide erroneous results in the presence of hidden nodes [Pearl, 2009, p. 54], measurement noise [Andersson, 2005, Nalatore et al., 2007] or improper preprocessing techniques [Gong et al., 2015]. However, we share the belief opined by Venkatesh et al. [2019] that the inability to interpret Granger causal influence as stimulus-related information flow is a much more fundamental issue, which limits the kinds of inferences one is able to make about the computation being performed by the neural circuit.

## 2 Results

The counterexample demonstrated by Venkatesh and Grover [2015] is based on a feedback communication scheme, which was originally proposed by Schalkwijk and Kailath [1966]. As mentioned before, while the counterexample was examined theoretically in a limited setting, it was never subjected to a computational evaluation or a rigorous statistical analysis. Our main results are two-fold:

1. We perform a rigorous statistical analysis of the feedback-based counterexample given by Venkatesh et al. [2019], and show that the result is much stronger than previously supposed: Granger causal influences can be statistically insignificant in the direction of information flow, while being highly significant in the opposite direction.

2. We also show that one can obtain a better understanding of the system by examining the stimulus-related information flow. In particular, by measuring the mutual and conditional mutual information between the signals and the stimulus—here, the message being communicated—allows one to interpret the true direction of information exchange in the system.

### 2.1 The Schalkwijk and Kailath Counterexample

The Schalkwijk and Kailath scheme [1966] is a strategy for efficiently communicating a message from a transmitter to a receiver (here, we refer to them as Alice and Bob respectively), in the presence of a feedback channel (see Fig. 1a). Suppose Alice wishes to send a message $\theta \sim \mathcal{N}(0, 1)$ to Bob. The feedforward and feedback channels between Alice and Bob are noisy, and have signal-to-noise ratios (SNRs) characterized by the noise variances, $\sigma_N^2$ and $\sigma_R^2$ respectively. To start with, we assume that the feedback channel has a *higher* SNR than the feedforward link[1], i.e., $\sigma_R^2 < \sigma_N^2$. The scheme proceeds iteratively: Alice starts by communicating the message to Bob, i.e., $X_1 = \theta$. Bob receives a noisy version, $Y_1 = X_1 + Z_1$, using which he computes an estimate $\hat{\theta}_1$. He transmits this estimate back to Alice on the feedback channel, and she receives the noisy estimate $\tilde{\theta}_1 = \hat{\theta}_1 + R_1$. Subsequently, Alice transmits the error in Bob's last best estimate: $X_i = \theta - \tilde{\theta}_{i-1}$; while Bob uses these noisy error terms to improve his estimate over time: $\hat{\theta}_i = \hat{\theta}_{i-1} + Y_i/i$. It can be shown that,

---

[1]The version of the scheme we present here is simplified from the original Schalkwijk and Kailath scheme, for ease of analysis. Also, this scheme is communication-theoretically optimal when the feedback channel is noiseless, however, it continues to work (if sub-optimally) even when noise is present in the feedback link.

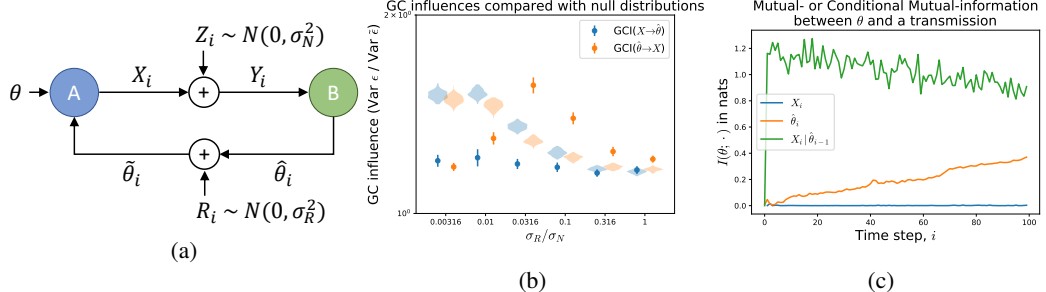

(a)

(b)

(c)

Figure 1: (a) A schematic of the counterexample based on the Schalkwijk-Kailath feedback communication scheme; (b) A comparison of Granger Causal influences (GCIs) at different reverse-noise-ratios, $\sigma_R/\sigma_N$. The violin plots indicate the null distributions based on the permutation test described in Section 2.1, while the errorbars show the mean and standard error of GCI. $\sigma_N = 0.1$ for this plot; (c) Mutual (and conditional mutual) information between the stimulus and Alice's and Bob's transmissions. $I(\theta; \hat{\theta}_i)$ slowly increases with $i$, while $I(\theta; X_i \mid \hat{\theta}_{i-1})$ slowly falls, indicating that Alice is communicating information about the message $\theta$ to Bob.

using this scheme, Bob's estimate eventually converges to the true value of the message: $\hat{\theta} \to \theta$ (see Venkatesh and Grover [2015] for a proof). Given the ubiquity of feedback links in the brain, such counterexamples deserve careful attention.

Suppose we now observe Alice's and Bob's transmissions ($X_i$ and $\hat{\theta}_i$), and wish to use a Granger causal analysis to determine how information flows in this setting. Intuitively, Alice's past transmissions do not predict Bob's future transmissions well: the $X_i$'s are corrupted by noise and $\hat{\theta}$ is a poor estimate initially. On the other hand, when the noise in the feedback link is small ($\sigma_R^2 < \sigma_N^2$), Bob's past transmissions predict Alice's future transmissions: $X_i = \theta - \tilde{\theta}_{i-1} \approx \theta - \hat{\theta}_{i-1}$. Since the Granger causal influence (GCI) from Alice to Bob effectively measures the extent to which Alice's past transmissions help in predicting Bob's future transmissions, we can conclude that the GCI from Bob to Alice is greater than that from Alice to Bob.

We demonstrate this computationally in Fig. 1b. We simulated the Schalkwijk and Kailath scheme for $T = 100$ time steps and for $n = 100$ trials. We computed GCIs by fitting an autoregressive model of order $p = 10$ to the data. Fig. 1b shows the mean GCI over 100 trials (errorbars represent standard error of the mean). We assessed the statistical significance of the result using the method described by Brovelli et al. [2004]: we permuted the trials of Alice's transmissions and Bob's transmissions independently, to disrupt trial-related dependences, while maintaining the original distributions of the individual transmissions. We then computed the GCIs on the permuted trials. We repeated this process $n_{\text{Perm}} = 100$ times, and constructed a histogram of mean GCIs under permutation, which became our empirical estimate of the null distribution. We found that for a certain regime of $\sigma_R/\sigma_N$, the actual GCI from Bob to Alice was far outside the empirical null distribution. The $p$-value of 0.01 was effectively the minimum attainable $p$-value, determined by the number of permutations we performed. Fig. 1b shows that GCIs can be statistically insignificant in the direction of information flow, while at the same time being highly significant in the opposite direction.

## 2.2 A Resolution through Mutual Information

Granger causality's failure to identify the direction in which the message flows in the above example can be attributed to the fact that Granger causality only examines predictive influence; it does not capture what that influence is *about*. Granger causality does not intrinsically check for stimulus-dependence in any way. The recent work of Venkatesh et al. [2019], while defining stimulus-related information flow, does not provide a quantitative measure of information flow, and their partial resolution to the counterexample based on derived information is cumbersome and unsatisfactory.

Here, we take a much simpler approach and show that by measuring mutual and conditional mutual information, we can observe how information about the message evolves in Alice's and Bob's transmissions. Since all variables in this example are Gaussian, the mutual information between the message $\theta$ and any transmission $U$ can be written in terms of their correlation: $I(\theta; U) = -\frac{1}{2} \log(1 - \rho(\theta, U)^2)$, where the correlation $\rho(\theta, U)$ is readily estimated. Fig. 1c shows how the mutual (and conditional mutual) information of $X_i$ and $\hat{\theta}_i$ evolve over time steps, $i$. In particular,

observe that $I(\theta; \hat{\theta}_i)$ slowly increases over time $i$, while $I(\theta; X_i)$ is nearly zero. The conditional mutual information $I(\theta; X_i \mid \hat{\theta}_{i-1})$, however, is much larger and slowly decreases over time, indicating the presence of synergistic information about $\theta$ in the forward link, which decays as the estimate $\hat{\theta}$ improves.

The decrease of stimulus-related information in Alice's transmissions, and the corresponding increase in Bob's transmissions indicates that information about the stimulus is being conveyed from Alice to Bob and not vice versa. This also indicates that caution must be exercised in interpreting Granger causal influences as conveying stimulus-related information.

## 3 Conclusion

We significantly advanced on a previously proposed counterexample, showing that it is possible for Granger causal influence to be statistically insignificant in the direction of stimulus-related information flow, while being highly significant in the opposite direction. We also demonstrated that quantitative information-theoretic measures, which are finding heavy use in the analysis of artificial neural networks, can be particularly useful in enabling the correct interpretation of the direction of information flow.

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
