# OpenReview forum: "Differentiating Granger Causal Influence and Stimulus-Related Information Flow"
_NeurIPS.cc/2019/Workshop/Neuro_AI — Submitted to Real Neurons & Hidden Units @ NeurIPS 2019_

### Official Review · AnonReviewer3 · 2019-09-20
**This study does not differentiate Granger causality and information flow**

**Clarity:** 3

**Comment:**

See above. The margins for improvement are very limited in this case.

**Category:**

Not applicable

**Clarity Comment:**

The words can be read, but the paper is difficult to understand until one realizes that it's about solving a non existing problem, and doing it wrong on top of it.

**Evaluation:**

1: Very poor

**Importance:**

1: Irrelevant

**Importance Comment:**

This paper aims to differentiate Granger causality and stimulus related information flow.
The problem is ill-posed, making it either impossible to solve or non-existing.
The latter case would correspond to a difference in definition (reduction in surprise vs reduction in variance). The former case would correspond to saying that Granger causality does not measure "true" information or "true" causality. And this is a tautology.

Regardless of this, the implementation is wrong, see next section.

**Intersection:**

1: Very low

**Intersection Comment:**

There is no link with neuro nor with AI

**Rigor Comment:**

For gaussian variables Granger causality and Transfer entropy are equivalent, see

Barnett, L., Barrett, A. B., & Seth, A. K. (2009). Granger Causality and Transfer Entropy Are Equivalent for Gaussian Variables. Physical Review Letters, 103(23). doi:10.1103/physrevlett.103.238701,

so the result of applying both algorithms to the case study presented here would be the same.

The results don't look the same because the authors compare two different things, in figure 1b is GC vs noise, in figure 1c is the conditioned mutual information vs time step.

Also, both results are correct. By construction there is a bidirectional influence (the feedback).

**Technical Rigor:**

1: Not convincing

---

### Official Review · AnonReviewer2 · 2019-09-24
**Mutual information complements Granger causal influence as a way to study information flow**

**Clarity:** 2

**Comment:**

As previously mentioned it will be increasingly important to develop these types of tools for both AI and neuroscience, but I feel the work is not at that point yet. The Schalkwijk and Kailath counterexample is a great one to build intuition, but application of these ideas to, say, and RNN trained on a neuroscience task would be a good addition to understand the differences between GCI and mutual information in a more relevant setting.

**Category:**

Common question to both AI & Neuro

**Clarity Comment:**

The authors did a reasonable job of explaining the problem and their approach given the limited space. I would like to hear more about what the authors think GCI is actually capturing, and how this is different than mutual information (in a context that is more general than the example given).

A small comment: the axis/figure text in Figure 1 is way too small to read.

**Evaluation:**

2: Poor

**Importance:**

4: Very important

**Importance Comment:**

Granger causality has been used in the fMRI literature for many years, and a deeper understanding of it's strengths and weaknesses, as well as complementary methods, will be increasingly important to neuroscience as other technologies for acquiring brain-wide activity come online.

**Intersection:**

1: Very low

**Intersection Comment:**

Neither Granger causality nor mutual information are what I would consider AI techniques. Though both can be used to understand deep networks and neuroscientific data, neither of those applications are presented here.

**Rigor Comment:**

The authors compare and contrast several well-known techniques, but the problem as stated was not entirely understandable to me.

**Technical Rigor:**

2: Marginally convincing

---

### Official Review · AnonReviewer1 · 2019-09-26
**An exploration of Granger Causality and Transfer Entropy in a Schalkwijk - Kailath model where those two values are the same thing.**

**Clarity:** 3

**Comment:**

This is interesting in its own right, but not moreso than a standard exercise in understanding TE/GC evolution on an error-correcting code system. It doesn't fit the bill for this workshop.

**Category:**

Common question to both AI & Neuro

**Clarity Comment:**

The body of text is itself readable, but the overall objective is confusing given the faults in the publication's premise. The figure is also hard to interpret -- the axes labels and figure legends are too small to be read clearly from a letter-/A4-sized representation of the paper.

**Evaluation:**

2: Poor

**Importance:**

2: Marginally important

**Importance Comment:**

Studying the causal influence of external stimuli and measuring the propagation of resulting information respresentations throughout neuobiological and artficial neural network models is a very important research topic, and a publication exploring that space of questions would be important if the systems studied therein were models relevant to the two use cases listed above; however, the results from this experiment do not add much of an understanding beyond what is already known in S-K systems.

**Intersection:**

2: Low

**Intersection Comment:**

While the model presented and studied here may be a valid model of some neural structures in its graph form, the model itself is too simplistic for this reviewer to consider it an intersection of AI and neuroscience. Granger Causality and Transfer entropy are widely used in both AI and neuroscience and their interpretations require further scrutiny and research, but I would not say that the work presented here is highly indicative of the implied intersection of the two fields given its relative triviality.

**Rigor Comment:**

The technical content of this publication is sound, but the context is not terribly relevant. As other reviewers have pointed out, it is known that transfer entropy and granger causality are equivalent measurements of gaussian signals in autoregressive systems, a set of models to which the Schalkwijk - Kailath error correction models belong. GC or TE computed between any given pair of signals in the system will be equal, so this study is not able to address the differences between the two measurements. All signals present are either explicitly iid (in time) sampled from respective gaussian distributions or are the difference between gaussians, which implicitly defines said difference signals as gaussian themselves.

Within-metric differences in GC and information-theoretic measurements are presented in figure 1(b) and 1(c); these show the evolution of information flow across time, but do nothing to highlight the presumed differences between GC and TE within the system presented and simulated. The results appear sound and consistent with this reviewer's general understanding of asymptotically correct systems, but they do not support the thesis presented by the paper, which itself appears to be a misunderstanding of the metrics considered.

**Technical Rigor:**

2: Marginally convincing

---

### Decision · Program_Chairs · 2019-10-01

**Decision:**

Reject

**Comment:**

Unfortunately, we had more submissions than we could accept and based on the review process, we have decided not to accept your submission.  Nevertheless, thank you for your submission and interest in our workshop.